# MPT: Multi-grained Prompt Tuning for Text-Video Retrieval

Haonan Zhang
University of Electronic Science and
Technology of China
China
zchiowal@gmail.com

Pengpeng Zeng*
Shenzhen Institute for Advanced
Study
University of Electronic Science and
Technology of China
China
is.pengpengzeng@gmail.com

Lianli Gao
Shenzhen Institute for Advanced
Study
University of Electronic Science and
Technology of China
China
lianli.gao@uestc.edu.cn

Jingkuan Song
Shenzhen Institute for Advanced
Study
University of Electronic Science and
Technology of China
China
jingkuan.song@gmail.com

Heng Tao Shen
University of Electronic Science and
Technology of China
Tongji University
China
shenhengtao@hotmail.com

## ABSTRACT

Recently, significant advancements have been made in supporting text-video retrieval by transferring large-scale image-text pretraining models through model adaptation, *i.e.*, full fine-tuning, or prompt tuning, a parameter-efficient fine-tuning strategy. While full fine-tuning involves high computational costs, particularly with increasing model size, prompt tuning offers greater flexibility and efficiency by adjusting only a few learnable parameters. However, current prompt tuning methods rely on coarse visual and textual cues for text-video retrieval task, neglecting the domain-specific features when performing the adaptation. This approach may lead to sub-optimal performance due to the incorporation of irrelevant and indiscriminate knowledge. To address such an issue, we present a **M**ulti-grained **P**rompt **T**uning (**MPT**) for text-video retrieval, that designs a variety of specific prompts to effectively explore semantic interaction across different modalities with diverse granularity. Specifically, we devise a multi-grained video encoder that employs spatial, temporal, and global prompts to transfer the base-generic knowledge from the image-text pretrained model while comprehensively excavating determinative video-specific characteristics. Meanwhile, we introduce a novel multi-grained text encoder aimed at capturing various levels of textual clues through the utilization of word and phrase prompts. Extensive experiments on four benchmark datasets, *i.e.*, MSR-VTT, ActivityNet, DiDeMo, and LSMDC, demonstrate that MPT achieves outstanding performance, surpassing state-of-the-art methods with

negligible computational cost. The codebase is publicly available at: https://github.com/zchoi/MPT.

## CCS CONCEPTS

• **Computing methodologies** → Visual content-based indexing and retrieval; **Image and video acquisition**.

## KEYWORDS

Text-video Retrieval, Prompt Tuning, Cross-modal Understanding, Vision-language Pre-training.

### ACM Reference Format:
Haonan Zhang, Pengpeng Zeng, Lianli Gao, Jingkuan Song, and Heng Tao Shen. 2024. MPT: Multi-grained Prompt Tuning for Text-Video Retrieval. In *Proceedings of the 32nd ACM International Conference on Multimedia (MM '24), October 28-November 1, 2024, Melbourne, VIC, AustraliaProceedings of the 32nd ACM International Conference on Multimedia (MM'24), October 28-November 1, 2024, Melbourne, Australia.* ACM, New York, NY, USA, 9 pages. https://doi.org/10.1145/3664647.3680839

## 1 INTRODUCTION

The field of multi-modal analysis has sparked tremendous interest with the explosion of multi-modal data and the growing power of deep learning methods. Among these, cross-modal alignment stands out as a crucial problem, necessitating an integrated comprehension and investigation of visual and language modalities. As a typical application of cross-modal learning, the text-video retrieval task [28, 49] aims to retrieve the most relevant video based on a text query, which is beneficial to numerous visual-language tasks, such as video captioning [41, 48], video question answering [36, 37], and video grounding [12, 22]. The general approach to this task follows the paradigm by first extracting video and text features separately with distinct backbones and then aligning those features into a common space. To date, it remains challenging due to the inherent heterogeneity gap between visual and textual modalities.

Large-scale image-text pre-trained models, *e.g.,* CLIP [34], a cornerstone invention in deep learning, have proven indispensable in multi-modal analysis. Benefiting from its powerful transferring

---

*Corresponding author.

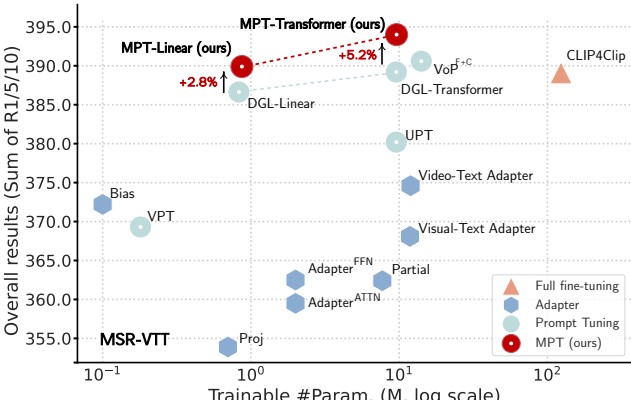

**Figure 1: Comparison of the proposed MPT with existing state-of-the-art methods in terms of efficacy (sum of R@1,5,10) and efficiency (#Param.) for text-video retrieval. Note that for a fair comparison, we showcase the results trained on the MSR-VTT dataset and all models adopt the CLIP-ViT-B/32 backbone.**

abilities, fine-tuning techniques have become prevalent to swiftly adapt these models to specific downstream tasks including image recognition [31] and video understanding [26]. As a precursor work adapts the large-scale image-text pre-trained models into text-video retrieval task, CLIP4Clip [28] successfully extends the above strategy that proposes a temporal fusion module to capture the dynamic features of different video frames and performs the cross-modal alignment on video and text features by a simple meaning pooling. However, the conventional paradigm of this full fine-tuning strategy often involves tremendous computational costs as the model size grows (Fig. 1), necessitating the storage of a separate set of model parameters for each dataset, thereby making practical deployment and scalability difficult. Moreover, fully fine-tuning the entire backbone also poses a risk of catastrophic forgetting of rich prior knowledge embedded in weights during the pre-training stage.

To alleviate these problems, there have been increasing efforts [5, 18, 24, 44] to explore *parameter-efficient* transfer learning for swift adaptions to downstream tasks such as text-video retrieval, *i.e.*, a flexible alternative tuning strategy that seeks to representative prior from the large-scale pre-trained image-text model by optimizing a set of learnable vectors, while keeping the pre-trained weights frozen. Two pioneering parameter-efficient strategies are Adapter tuning [17, 39] and Prompt tuning [19, 24]. Specifically, Adapter tuning requires fine-tuning partial model parameters, *e.g.*, BIAS [9], PROJ [19], and PARTIAL [19], or injects a learnable module into the model, *e.g.*, Adapter$^{\text{ATTN/FFN}}$ [11, 16]. However, the adapter-based approaches shown in Fig. 1 have limited retrieval performance and often require the modification of the model backbone to insert the external blocks. Compared with it, prompt tuning only learns a small set of learnable vectors (*called prompts*) without changing other model parameters. It obtains performance on par with or outperforms the full fine-tuning counterparts while efficiently reducing the trainable parameters, as shown in Fig. 1. For instance, VoP [18] first introduces the prompt tuning into text-video retrieval

and designs three kinds of video-specific prompts while optimizing the dual branches' prompts independently. DGL [44] proposes a cross-modal dynamic prompt tuning method aiming to capture global video information and encourage inter-modal interaction.

Despite the significant promotion, current prompt tuning methods solely concentrate on coarse-grained interaction for text-video retrieval, *i.e.*, only one type of prompt for the corresponding modality, which hampers the capture of the domain-specific features when performing the adaptation. Essentially, the videos possess various inherent properties that contribute to their richness and complexity such as spatial, temporal, and global information. In addition, the text also embodies a variety of intrinsic properties including lexical and contextual semantics, where the former emphasizes the word-level meaning and the latter controls the associations between joint words, *e.g.*, under different contexts, the meaning of single word 'teddy' in '*Teddy bear*' and '*President Teddy*' is completely different. Therefore, it is essential to analyze the multi-grained semantics in video-text modalities for prompt tuning-based text-video retrieval tasks, which also greatly benefits the practical retrieval system.

Based on the above insights, we propose Multi-grained Prompt Tuning (MPT) for text-video retrieval. Our core idea is to transfer the encyclopedic knowledge from the pre-trained model while excavating the multi-grained domain-specific features by leveraging several distinct prompt vectors. To be specific, we first introduce a multi-grained video encoder (MVE) that incorporates three types of prompts, *i.e.*, spatial prompts, temporal prompts, and global prompts to learn the video-specific characteristics in a local-to-global manner thoroughly. Likewise, we symmetrically propose a multi-grained text encoder (MTE) by introducing the word prompts and phrase prompts to capture the intrinsic lexical and contextual semantics of the given sentence. Note that the phrase features are obtained from the word embeddings via a prototype-based learning strategy without any supervision. Moreover, the above prompting processes are in parallel and parameter-shared within each layer for both video and text encoders to effectively maintain the model size. We carry out extensive experiments on four commonly used text-video datasets including MSR-VTT, ActivityNet, DiDeMo, and LSMDC. Learning multi-grained video and text semantics while maintaining powerful pre-trained knowledge, our MPT consistently achieves state-of-the-art performance compared to previous methods. In summary, the main contributions of our work are three-fold:

- We propose a multi-grained prompt tuning (MPT) for text-video retrieval, which endows the pre-trained image-text model with multiple fine-grained considerations of video-text modalities, thus facilitating the excavations of the domain-specific features.
- We introduce a multi-grained video encoder by incorporating three levels of prompts, *i.e.*, spatial, temporal, and global prompts, to learn the inherent properties of the video modality. Accordingly, a multi-grained text encoder is devised to model the lexical and contextual semantics of the given sentence via a word prompt and a phrase prompt.
- Through extensive experiments on four commonly used text-video benchmarks, *i.e.*, MSR-VTT, ActivityNet, DiDeMo, and LSMDC, we demonstrate that MPT significantly outperforms previous methods, achieving the best trade-off between performance and computational cost.

  

## 2 RELATED WORKS

### 2.1 Vision-Language Pre-training

Vision-Language Pre-training (VLP) aims to learn joint representations of vision and language, thereby outperforming across a variety of downstream tasks. Recently, profiting from large-scale visual and textual pairs collected from the Internet, the contrastive text-image pre-training [6, 29, 34, 40, 46] gains significant achievements. As a pioneering work, CLIP (Contrastive Language-Image Pre-training) [34] utilizes a contrastive loss to train two uni-modal encoders via 400 million pairs of images and texts. And the success of derivative works also highlights the adaptability of pre-trained models such as Flamingo [2] and ALBEF [25]. For video counterparts, most video-language pre-training [4, 23] is based on large-scale text-video datasets such as HowTo100M [30] and WebVid-2M [4]. Despite the advent of works like CLIPBERT [23] and Frozen in Time [4] which have demonstrated considerable potential in video-language understanding tasks, there are still challenges that need to be overcome. It is worth noting that videos present challenges in terms of acquisition costs and effort, leading to limited scale and suboptimal generalization capabilities. Moreover, video-language pre-training also requires substantial computing resources and contends with significant text-irrelevant noise interference in video data. To mitigate this burden, models like CLIP4Clip [28] and X-Pool [15] are proposed to transfer the powerful generalization ability in image-text pre-training to the video domain. Accordingly, our work follows this scheme for text-video retrieval.

### 2.2 Prompt Learning

Prompt learning, stemming from language processing (NLP), aims to adapt pre-trained language models to various downstream tasks. The original prompts are meticulously crafted language templates that necessitate substantial expertise and are limited in their ability to generalize. In order to address the aforementioned issues, the researchers suggest implementing *prompt tuning*, a technique that involves introducing learnable tokens as prompts and exclusively optimizing these tokens throughout the training process. Motivated by the achievements of prompt learning in NLP, this paradigm is extended to vision language models (VLMs). For example, CoOp [50] applies a series of learnable text prompts into the text input, which is jointly optimized with image labels during the training process to improve classification accuracy. Unlike adding prompts on the text side, VPT [19] employs token-level or pixel-level prompts to the vision branch, aiming to grasp the intrinsic attributes and relationships in images. However, the works above consider only the vision or text-side prompting, which ignores the synergy between the two modalities. MaPLe [21] introduces multi-modal prompts, breaking the isolation of visual and text prompts. Recently, the application of prompt learning in VLMs extends beyond image processing to video-related tasks like video understanding and text-video retrieval. For example, VoP [18] makes innovations in visual prompts and designs different ways to capture the spatio-temporal characteristics of videos. DGL [44] achieves the modal interaction of video and text prompts and utilizes global-local prompts interaction to capture global video information. In this work, we further mine more fine-grained information of both modalities to enhance the performance of text-video retrieval.

### 2.3 Text-video Retrieval

Text-video retrieval aims to determine and rank videos based on their semantically correspondent textual queries and *vice versa*. Previous works like [7, 13, 38, 51] have largely concentrated on complex fusion modules after extracting offline features, attempting to map pre-processed text and video data into a common latent space for alignment. In recent years, significant improvements have been achieved by adapting large-scale pre-trained image-text models such as CLIP [34] to various downstream tasks, *e.g.*, image classification [1, 33], video localization [10, 43], and visual question answering (VQA) [32, 37]. This paradigm has encouraged researchers to adapt these robust pre-trained image-text models for text-video retrieval, yielding unprecedented performance. As a preliminary study, CLIP4Clip [28] fine-tunes the pre-trained CLIP model to text-video retrieval with several similarity calculations, achieving remarkable performance on various benchmarks. X-Pool [15] highlights the deficiency of text-agnostic video pooling and thus proposes a text-conditioned video interaction for text-video retrieval. DicoSA [20] devices a disentangled conceptual framework and aligns video sets to simulate human reasoning processes. In contrast to the methods described previously, the present work seeks to delve into the multi-grained interactions between video and text using a more parameter-efficient strategy for text-video retrieval.

## 3 METHOD

In this section, we present our Multi-grained Prompt Tuning (MPT) for text-video retrieval. Specifically, we begin with a brief overview of some critical preliminaries including the task formulation of text-video retrieval, CLIP-based paradigm, and prompt tuning in Sec. 3.1. Then, we provide a detailed exposition of MPT in Sec. 3.2, containing two novel components, *i.e.*, multi-grained video encoder (MVE) and multi-grained text encoder (MTE), to acquire multi-grained properties from both visual and textual representations. At last, we describe the objective function for text-video retrieval in Sec. 3.3. The overall framework of MPT is depicted in Fig. 2.

### 3.1 Preliminary

**Task Formulation.** The text-video retrieval task seeks to explore well-aligned representations between the texts and videos, including two sub-tasks, namely text-to-video retrieval (*t2v*) and video-to-text retrieval (*v2t*). In *t2v*, it aims to retrieve the semantic-related video $v \in \mathcal{V}$ based on the text query $t \in \mathcal{T}$, and *vice versa* for *v2t*, where $\mathcal{V}$ and $\mathcal{T}$ indicate a video gallery and a text gallery, respectively. In practice, a dual-branch architecture is usually adopted, where the video and text features are first extracted using a TextEncoder and a VideoEncoder, respectively, and then a similarity function $s$ is to measure the semantic relevance between them, which can be formalized as:

$$
\begin{aligned}
z_t &= VideoEncoder(t), \\
z_v &= TextEncoder(v), \\
s(t,v) &= \frac{\mathbf{z}_t \cdot \mathbf{z}_v}{\|\mathbf{z}_t\|\|\mathbf{z}_v\|}.
\end{aligned}
\tag{1}
$$

**Revisiting CLIP-based Paradigm.** As a dominant vision-language model pre-trained on massive web-scale image-text pairs, CLIP [34]

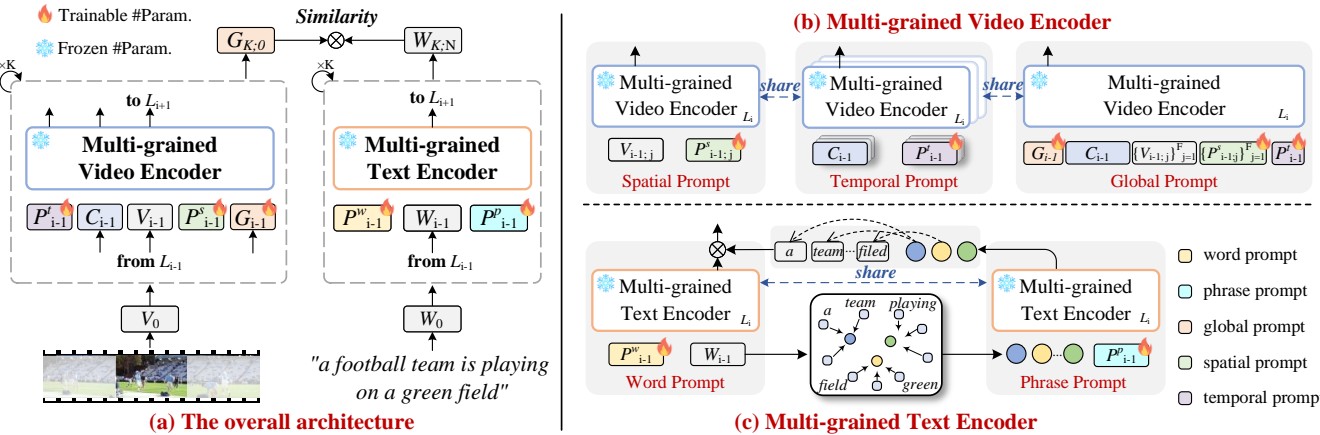

**Figure 2: Overview of the proposed Multi-grained Prompt Tuning (MPT) for text-video retrieval. The left side showcases: (a) the overall architecture of the proposed method. It learns the domain-specific features when performing pre-trained adaptation by involving two novel components: (b) a multi-grained video encoder that leverages three types of prompts, *i.e.*, spatial, temporal, and global, to acquire the inherent video characteristics in parallel, and (c) a multi-grained text encoder to model the intrinsic lexical and contextual semantics of the description by involving word prompts and phrase prompts.**

has experimented with significant advancements in generalizing to various downstream tasks such as image classification [1, 33] and video localization [10, 43]. Owing to its dual-branch structure, CLIP becomes a feasible and effective solution to learning cross-modal representation for text-video retrieval.

**(1) Text Encoder.** To extract text features, we first tokenize and pad the input sentence $t$ into a fixed-length token sequence and then project it into a word embedding $W_0 \in \mathbb{R}^{N \times D_t}$, where $N$ is the number of the text tokens and $D_t$ is the textual embedding size. Then the text features are learned by sending the word embeddings into a $K$-layer Transformer of pre-trained CLIP, where the operation of $\mathcal{F}^t$ is defined as:

$$W_i = \mathcal{F}_i^t(W_{i-1}). \tag{2}$$

Finally, we take the embedding of the [EOS] token $W_K^N$ from the last layer of the text encoder as the sentence embedding $z_t$:

$$z_t = \text{TextProj}(W_K^N). \tag{3}$$

**(2) Video Encoder.** Correspondingly, to extract video features, we first split each frame of the input video $v$ into $M$ non-overlapping patches and then project them into a frame sequence $V_0 \in \mathbb{R}^{F \times M \times D_v}$, where $F$ is the number of video frames and $D_v$ is the visual embedding size. Then, we learn the frame features by continually inputting each video frame $V_{i;j} \in \mathbb{R}^{M \times D_v}$ appended with a [CLS] tokens $c_{i;j} \in \mathbb{R}^{D_v}$ into the image encoder $\mathcal{F}^v$ of the pre-trained CLIP. Usually, each [CLS] token will learn the global information about the corresponding video frame. Giving $i$-th layer of the video encoder:

$$[c_{i;j}, V_{i;j}] = \mathcal{F}_i^v([c_{i-1;j}, V_{i-1;j}]), \tag{4}$$

where $[,]$ indicates the concatenate operation. By selecting all [CLS] tokens from the last layer as the frame embeddings, we obtain the video embedding as follows:

$$z_v = \text{VisualProj}(c_K). \tag{5}$$

**Prompt Tuning.** Despite the progress made in CLIP-based full fine-tuning algorithm, this strategy inevitably involves computational costs as it requires updating all parameters, making it infeasible to adapt to various datasets or deploy on practical retrieval systems. To move a step further for a more effective and efficient retrieval model, prior methods have adopted prompt tuning to text-video retrieval task which has shown great progress. A basic formulation of prompt turning for this task is summarized as follows:

$$\begin{aligned} [W_i, \_] &= \mathcal{F}_i^t([W_{i-1}, T_{i-1}]), \\ [c_{i;j}, V_{i;j}, \_] &= \mathcal{F}_i^v([c_{i-1;j}, V_{i-1;j}, E_{i-1}]), \end{aligned} \tag{6}$$

where $T_{i-1} \in \mathbb{R}^{H_t \times D_v}$ indicates the textual prompt and $E_{i-1} \in \mathbb{R}^{H_v \times D_v}$ indicates the visual prompt for the $i$-th layer, respectively. '$\_$' denotes the outputs at the corresponding positions are discarded. $H_t$ and $H_v$ mean the length of the textual and visual prompts. Nonetheless, current prompt tuning methods are limited to coarse-grained interaction for text-video retrieval. They typically utilize only one type of prompt for each modality, which restricts their ability to capture domain-specific features during adaptation. To this end, we seek to learn the multi-grained multi-modal semantics of both video and text via prompt tuning.

## 3.2 Multi-grained Prompt Tuning (MPT)

In this work, we propose a Multi-grained Prompt Tuning (MPT) for text-video retrieval task by incorporating different levels of prompts to capture various properties of video and text modalities. **Multi-grained Video Encoder.** As discussed in Sec. 1, video inherently contains multiple visual properties, *i.e.*, static spatiality (intra-frame), dynamic temporality (inter-frame), and global understanding (whole video). However, solely adopting one type of visual prompt like Eq. 6 ignores the above-mentioned video characteristics, which may lead to sub-optimal retrieval performance. To alleviate this issue, we introduce a multi-grained video encoder (MVE) that learns three types of prompts, *i.e.*, spatial, temporal, and

global prompts, to acquire adequate video-specific semantics, as shown in Fig. 2 (b).

For the **spatial prompts**, we remove the [CLS] tokens from Eq. 6 to exclusively learn the pure intra-frame patch information. By doing this, the flow of each video encoder layer in Eq. 4 becomes:

$$[V_{i;j}, \_] = \mathcal{F}_i^v([V_{i-1;j}, P_{i-1;j}^s]), \tag{7}$$

where $P_{i-1;j}^s \in \mathbb{R}^{H_s \times D_v}$ is the spatial prompts for the $i$-th layer of the $j$-th frame, and $H_s$ is the length of the spatial prompts. Note that we leave the [CLS] token in the subsequent temporal learning and restrict the spatial prompts to be shared between all frames that are at the same relative position in their respective videos.

For the **temporal prompts**, we encourage the dynamic correlation between consecutive video frames by considering the [CLS] token-based learning, which is typically designed to contain the general information of video frames. In parallel with spatial prompting, we combine the temporal prompts with [CLS] tokens of all frames together as the input for each video encoder layer:

$$[c_i, \_] = \mathcal{F}_i^v([c_{i-1}, P_{i-1}^t]), \tag{8}$$

where $P_{i-1}^t \in \mathbb{R}^{H_t \times D_v}$ is the temporal prompts belong to $i$-th layer and $H_t$ is the length of the temporal prompts.

For the **global prompts**, it is mainly responsible for getting the overall information of the video across all spatial-temporal dimensions. Inspired by [44], the global prompts are acting like a Q-former mechanism to extract the most useful visual information:

$$[G_i, \_, \_, \_] = \mathcal{F}_i^v([G_{i-1}, c_{i-1}, V_{i-1}, P_{i-1}^s, P_{i-1}^t]), \tag{9}$$

where $G_{i-1} \in \mathbb{R}^{H_g \times D_t}$ is the global prompts of the $i$-th layer and $H_g$ is the length of the global prompts. Different from the spatial and temporal prompts, we take the global prompts as the query and the others as the key and value for the multi-head attention mechanism to (1) comprehensively summarize the overall video information from other features, and (2) efficiently reduce the computational complexity. Note that the above three prompting processes are in parallel in each video encoder layer and each layer is parameter-shared between them to maintain the model size.

**Multi-grained Text Encoder.** Recent approaches [18, 44] mainly devote attention to the video branch by introducing various prompt tuning strategies, while ignoring the intrinsic language semantics. Intuitively, a deeper language understanding requires a lexical level learning of the main single words, and then forming them into phrases for further contextual comprehension. Motivation by this, we propose a multi-grained text encoder (MTE) by employing word and phrase prompts to learn textual clues from different aspects.

For the **word prompts**, we follow the basic formulation in Eq. 6 and simply prepend them with the text feature, forming the input of the $i$-th layer of the text encoder to capture lexical-level semantics of the word embeddings:

$$[W_i, \_] = \mathcal{F}_i^t([W_{i-1}, P_{i-1}^w]), \tag{10}$$

where $P_{i-1}^w \in \mathbb{R}^{H_w \times D_t}$ is the word prompts of the $i$-th layer of the text encoder and $H_w$ is the length of the word prompts.

For the **phrase prompts**, to obtain the phrase embeddings, we first introduce an unsupervised prototype-based strategy to group semantically related word embedding into a common centroid. Compared to the previous phrase-parsing algorithm, our method is more

time-saving and flexible. Particularly, within each text encoder layer, we define $R$ cluster centers $Z_i = \{Z_{i;r}\}_{r=1}^R \in \mathbb{R}^{R \times D_t}$ and utilize dot product to calculate the similarities between word embeddings with these centers. Given a word embedding, its assignments to the $r$-th cluster can be generated as follows:

$$a_{n,r} = \frac{exp(W_{i-1;n}Z_{i-1;r}^T + b_{i-1;r})}{\sum_{r'} exp(W_{i-1;n}Z_{i-1;r'}^T + b_{i-1;r'})}, \tag{11}$$

where $b_{i-1;n/o}$ is trainable parameters. Then we can obtain the aggregated residual features for each cluster as formulated below:

$$O_{i-1;r} = Norm(\sum_{r=1}^R a_{n,r}(W_{i-1;n} - \tilde{Z}_{i-1;r})), \tag{12}$$

where '$Norm$' means the $\ell_2$-normalization and $\tilde{Z}_{i-1;r}$ means the trainable weights with the same size as $Z_{i-1;r}$. To this end, we obtain a set of contextualized phrase features $O_{i-1} = \{O_{i-1;r}\}_{r=1}^R \in \mathbb{R}^{R \times D_t}$ for the $i$-th layer. Then, we combine generated phrase features with phrase prompt as the input for $i$-th text encoder layer:

$$[O_i, \_] = \mathcal{F}_i^t([O_{i-1}, P_{i-1}^p]). \tag{13}$$

where $P_{i-1}^p \in \mathbb{R}^{H_p \times D_t}$ is the phrase prompts and $H_p$ is the length of the phrase prompts. Finally, we back-add the cluster features to their corresponding word embeddings for semantic enhancement:

$$W_{i;n} = W_{i;n} + O_{i;r}, \quad if \ W_{i;n} \in O_{i;r}, \tag{14}$$

where the aggregated word embeddings $W_i$ are prepared for the input of the next layer of the textual encoder.

### 3.3 Objective Function

Following [44], we utilize contrastive loss to train our model, regarding the paired text-video data as positive and the others as negative in a mini-batch. We optimize the symmetric text-to-video and video-to-text losses as follows:

$$\begin{aligned} \mathcal{L}_{t2v} &= \frac{1}{B} \sum_{i=1}^B \log \frac{e^{s(z_t^i, z_v^i)/\tau}}{\sum_{j=1}^B e^{s(z_t^j, z_v^i)/\tau}}, \\ \mathcal{L}_{v2t} &= \frac{1}{B} \sum_{i=1}^B \log \frac{e^{s(z_t^i, z_v^i)/\tau}}{\sum_{j=1}^B e^{s(z_t^i, z_v^j)/\tau}}, \end{aligned} \tag{15}$$

where $B$ represents the mini-batch size, $\tau$ is a learnable temperature scale, and $s(z_t^i, z_v^j)$ is the cosine similarity between the text representation $z_t^j$ and the video representation $z_v^i$. To this, the final loss function is:

$$\mathcal{L}_{\text{retrieval}} = \frac{1}{2}(\mathcal{L}_{t2v} + \mathcal{L}_{v2t}). \tag{16}$$

Note that different from the Eq. 5, we follow [44] and take global prompt in first position $G_{K;0}$ from the last layer as video embedding.

## 4 EXPERIMENTS

### 4.1 Datasets and Evaluation Metrics

**Datasets.** To verify the effectiveness of our proposed method, we conduct extensive experiments on four widely used datasets for text-video retrieval, including MSR-VTT [42], ActivityNet [8], DiDeMo [3], LSMDC [35]. **MSR-VTT** contains 10,000 video clips,

**Table 1: Comparison with state-of-the-art on the MSR-VTT dataset. Here, along with the performance of common retrieval metrics, we also report the number of trainable parameters (#TP) and the sum of all recalls (SumR).**

| Types | Methods | #TP (M) | Text ⇒ Video | | | | Video ⇒ Text | | | | SumR ↑ |
|-------|---------|---------|------|------|-------|------|------|------|-------|------|--------|
| | | | R@1↑ | R@5↑ | R@10↑ | MnR↓ | R@1↑ | R@5↑ | R@10↑ | MnR↓ | |
| *CLIP-ViT-B/32* | | | | | | | | | | | |
| Finetune | CLIP4Clip [28] | 123.54 | 43.1 | 70.4 | 80.8 | 16.2 | 43.1 | 70.5 | 81.2 | 12.4 | 389.1 |
| Adapter | Bias [9] | 0.1 | 39.7 | 66.5 | 77.3 | 17.3 | 41.1 | 68.4 | 79.2 | 13.6 | 372.2 |
| | Proj [19] | 0.7 | 37.1 | 63.0 | 76.1 | 20.5 | 37.2 | 64.6 | 75.9 | 16.7 | 353.9 |
| | Partial [19] | 7.7 | 39.8 | 65.3 | 75.9 | 19.3 | 37.9 | 66.1 | 77.4 | 15.5 | 362.4 |
| | Adapter$^{ATTN}$ [16] | 2.0 | 37.6 | 63.2 | 75.8 | 18.7 | 39.6 | 66.5 | 76.8 | 14.7 | 359.5 |
| | Adapter$^{FFN}$ [11] | 2.0 | 38.2 | 63.5 | 76.4 | 17.9 | 39.9 | 66.8 | 77.7 | 14.2 | 362.5 |
| | Visual-Text Adapter [44] | 11.82 | 39.2 | 65.7 | 76.1 | 17.6 | 40.7 | 68.8 | 77.6 | 13.7 | 368.1 |
| | Video-Text Adapter [44] | 11.94 | 41.1 | 67.0 | 77.1 | 17.4 | 42.6 | 68.4 | 78.4 | 13.8 | 374.6 |
| Prompt | VPT [19] | 0.18 | 42.0 | 66.6 | 77.3 | 19.2 | 39.4 | 66.8 | 77.2 | 16.2 | 369.3 |
| | UPT [47] | 9.57 | 42.1 | 67.7 | 78.2 | 16.5 | 42.6 | 70.3 | 79.3 | 12.3 | 380.2 |
| | VoP$^{F+C}$ [18] | 14.10 | 44.6 | 69.9 | 80.3 | 16.3 | 44.5 | 70.7 | 80.6 | 11.5 | 390.6 |
| | DGL-Linear [44] | 0.83 | 44.7 | 70.5 | 79.2 | 16.2 | 42.1 | 70.0 | 80.6 | 13.4 | 387.1 |
| | DGL-Transformer [44] | 9.57 | 45.8 | 69.3 | 79.4 | 16.3 | 43.5 | 70.5 | 80.7 | 13.1 | 389.2 |
| | **MPT-Linear** (ours) | 0.87 | 45.0 | 70.8 | 79.6 | 16.2 | 42.8 | 70.6 | 81.1 | 12.9 | 389.9 |
| | **MPT-Transformer** (ours) | 9.61 | 46.3 | 70.9 | 80.7 | 15.6 | 45.0 | 70.9 | 80.6 | 12.7 | 394.4 |
| *CLIP-ViT-B/16* | | | | | | | | | | | |
| | CLIP4Clip [28] | 123.54 | 45.6 | 71.2 | 80.9 | 15.2 | 43.2 | 72.5 | 80.7 | 10.9 | 394.1 |
| | VoP$^{F+C}$ [18] | 14.10 | 47.7 | 72.4 | 82.2 | 12.0 | - | - | - | - | - |
| | DGL-Linear [44] | 0.83 | 48.3 | 71.8 | 80.6 | 13.4 | 45.7 | 74.0 | 82.9 | 10.9 | 403.3 |
| | DGL-Transformer [44] | 9.57 | 48.6 | 71.8 | 82.2 | 13.6 | 46.3 | 74.2 | 83.8 | 9.9 | 406.9 |
| | **MPT-Linear** (ours) | 0.87 | 48.3 | 72.0 | 81.7 | 14.9 | 46.5 | 74.1 | 82.6 | 11.8 | 405.2 |
| | **MPT-Transformer** (ours) | 9.61 | 49.2 | 72.9 | 82.4 | 15.5 | 47.4 | 73.9 | 83.4 | 10.9 | 409.3 |

each annotated with about 20 human-labeled descriptions. Following [15, 28, 44], we utilize "training-9K" split [14] for training and "test 1K-A" split [45] for testing, which includes 9,000 and 1,000 video-descriptions pairs, respectively. **ActivityNet** is a long-video dataset, which collects 20,000 videos with 200 different types of human activities from YouTube. Following [20], we concatenate all of the video descriptions into a paragraph to test the model with video-paragraph retrieval on the "val1" split. **DiDeMo** has 10,000 Flikr videos described by 40,000 sentences. Like the ActivityNet dataset, all descriptions of a video are merged into a query to evaluate the model. **LSMDC** is composed of 118,081 video clips extracted from 202 movies, each of which has a single caption. There are 109,673, 7,408, and 1,000 videos for training, validation, and testing. **Evaluation Metrics.** For a fair comparison, we follow the existing works [18, 44] and employ standard retrieval metrics to evaluate the performance of the proposed model, including R@K (Recall at Rank K, higher is better ↑) and MnR (Mean Rank, lower is better ↓). Specifically, R@K measures the percentage of ground-truth hits in the top-K ranking list. Here, K is set to 1, 5, and 10, respectively.

## 4.2 Implementation Details

Following existing works [18, 44], the video and text encoders are initialized with the pre-trained CLIP [34], and all pre-trained weights are frozen during the model training. The visual and textual embedding sizes $D_v$ and $D_t$ are set to 768 and 512 respectively. For the initialization of prompts, inspired by UPT [47], we adopt a

shared embedded encoding layer to generate the word&phrase and spatial&temporal&global prompts, which can better enhance the interaction across modalities compared to generating different prompts solely. Moreover, we further encode these prompts by utilizing one linear layer or one transformer layer to build the relationship among them, resulting in two versions of our model, *i.e.*, **MPT-Linear** and **MPT-Transformer**. By default, the length of all the above prompts is set to $H_{s/t/g/w/p} = 4$. For MSR-VTT and LSMDC datasets, we set the max length of the caption to $N = 32$ and uniformly sample $F = 12$ frames per video. For DiDeMo and ActivityNet datasets, the maximum number of sentences and frames is all set to $N/F = 64$. All video frames are resized to $224 \times 224$ and split into $M = 49$ non-overlapping patches. The cluster number is set to $R = 5$ to learn the phrase features. During training, the model is trained within 10 epochs by the AdamW optimizer with 0.2 decoupled weight decay. The initial learning rate is set to $1e - 2$ and $5e - 3$ for LSMDC and other datasets, respectively, with warm up by a cosine scheme [27]. The mini-batch size is 128 for MSR-VTT and LSMDC and 64 for DiDeMo and ActivityNet, respectively. All experiments are carried out on 8 NVIDIA RTX A6000 GPUs.

## 4.3 Performance Comparisons

**Compared Methods.** In this section, to evaluate the capability of the proposed MPT, we compare it with state-of-the-art methods, where all methods are built on the CLIP backbone. The comparison methods can be briefly classified into three categories: **(i)**

**Table 2: Comparison with state-of-the-art on the ActivityNet, DiDeMo, and LSMDC. Briefly, we only report text-to-video retrieval (*t2v*) results. * denotes that we evaluated performances using public code provided by corresponding papers.**

| Types | Methods | ActivityNet | | | | DiDeMo | | | | LSMDC | | | |
|---|---|---|---|---|---|---|---|---|---|---|---|---|---|
| | | R@1↑ | R@5↑ | R@10↑ | MnR↓ | R@1↑ | R@5↑ | R@10↑ | MnR↓ | R@1↑ | R@5↑ | R@10↑ | MnR↓ |
| Finetune | CLIP4Clip [28] | 40.5 | 72.4 | 98.1 | 7.5 | 43.4 | 70.2 | 80.6 | 17.5 | 20.7 | 38.9 | 47.2 | 65.3 |
| Adapter | Bias [9] | 31.3 | 60.3 | 74.2 | 13.4 | 36.5 | 63.4 | 75.2 | 24.8 | 17.4 | 36.2 | 44.9 | 73.2 |
| | Proj [19] | 29.8 | 59.1 | 73.3 | 14.2 | 35.6 | 61.3 | 72.6 | 24.4 | 15.7 | 32.7 | 40.8 | 83.7 |
| | Partial [19] | 33.6 | 64.0 | 77.8 | 10.6 | 39.3 | 65.5 | 75.7 | 22.3 | 18.0 | 33.8 | 41.8 | 79.9 |
| | Adapter$^{\text{ATTN}}$ [16] | 31.6 | 60.5 | 74.4 | 13.1 | 36.4 | 62.8 | 73.9 | 23.5 | 18.4 | 38.0 | 46.4 | 68.9 |
| | Adapter$^{\text{FFN}}$ [11] | 31.8 | 61.0 | 75.0 | 12.8 | 36.3 | 63.4 | 75.4 | 22.9 | 18.7 | 38.9 | 47.3 | 63.6 |
| | Visual-Text Adapter [44] | 33.5 | 64.8 | 77.5 | 10.9 | - | - | - | - | 18.0 | 34.4 | 43.5 | 75.2 |
| | Video-Text Adapter [44] | 36.4 | 66.1 | 79.6 | 10.0 | - | - | - | - | 18.3 | 35.5 | 44.0 | 74.8 |
| Prompt | VoP$^{\text{F+P}}$ [18] | 36.1 | 65.5 | 78.5 | 10.9 | 45.3 | 72.3 | 80.4 | 13.8 | 20.7 | 40.7 | 49.7 | 59.1 |
| | DGL-Transformer [44] | 40.1 | 69.5 | 80.9 | 9.1 | 45.6* | 71.7* | 81.1* | 14.6* | 21.2 | 37.8 | 48.8 | 66.5 |
| | **MPT-Transformer** (ours) | 41.4 | 70.9 | 82.9 | 7.8 | 46.4 | 72.2 | 81.4 | 13.4 | 21.1 | 41.2 | 49.4 | 63.2 |

**Table 3: Ablation study of integrating different prompts in MPT. MVE and MTE indicate the multi-grained video/text encoders. S.P: spatial prompts. T.P: temporal prompts. G.P: global prompts. W.P: word prompts. P.P: phrase prompts.**

| Line | MVE | | | MTE | | Text ⇒ Video | | | |
|---|---|---|---|---|---|---|---|---|---|
| | S.P | T.P | G.P | W.P | P.P | R@1↑ | R@5↑ | R@10↑ | MnR↓ |
| 1 | ✓ | ✗ | ✗ | ✓ | ✗ | 42.0 | 67.5 | 77.6 | 16.4 |
| 2 | ✓ | ✓ | ✗ | ✓ | ✗ | 44.0 | 70.2 | 79.6 | 15.9 |
| 3 | ✓ | ✓ | ✓ | ✓ | ✗ | 45.3 | 70.2 | 80.3 | 16.2 |
| 4 | ✓ | ✓ | ✓ | ✓ | ✓ | 46.3 | 70.9 | 80.7 | 15.6 |

**Finetune**: updating the entire model's parameters during training, including CLIP4Clip [28]. **(ii) Adapter**: updating only selected parameters of the model, or additional lightweight learnable modules inserted into the model, including Bias [9], Proj [19], Partial [19], Adapter$^{\text{ATTN}}$ [16], Adapter$^{\text{FFN}}$ [11], Visual-Text Adapter [44], and Video-Text Adapter [44]. **(iii) Prompt**: updating only a few additional learnable prompt tokens prepended to input tokens and keeping backbone frozen, including VPT [19], UPT [47], VoP$^{\text{F+C}}$ [18], and DGL [44]. Our proposed MPT belongs to the third one.

**Comparisons on MSR-VTT dataset.** Tab. 1 presents the performance comparison on the MSR-VTT dataset under the setting of two backbones, *i.e.*, CLIP-ViT-B/32 and CLIP-ViT-B/16. In addition, to better showcase the efficacy and efficiency, we also report the trainable parameters (#TP) and the sum of recall@1/5/10 (SumR). From the table, we observe that: (1) Compared with the existing methods (ViT-B/32), our MPT-Transformer obtains an obvious gain on most evaluation metrics. In particular, our method outperforms the best counterpart VoP$^{\text{F+C}}$ on the SumR metric by large margins of 5.2%, which indicates overall high-quality retrieval results of our method. Besides, it achieves better performance in terms of R@1 than the DGL-Transformer without excessive parameters, increased by 0.5% and 1.5% for text-to-video and video-to-text retrieval, respectively. (2) During training, both two versions of our method update only 0.7% and 7.7% of the parameters than the fully fine-tuned method, *i.e.*, CLIP4Clip. Compared to parameter-efficient

methods, although MPT may not have the least trainable parameters, it strikes a better trade-off between performance and efficiency. (3) By adopting a more powerful backbone, *i.e.*, CLIP-ViT-B/16, our proposed MPT yields a further performance improvement, which consistently surpasses previous fine-tuning and prompt tuning methods. The above results clearly prove the validity of MPT.

**Results on other Datasets.** To verify the robustness of our method, we further provide quantitative experiments on the other three datasets, *i.e.*, ActivityNet, DiDeMo, and LSMDC, in Tab. 2, where the table only reports text-to-video results (*t2v*) for simplicity. We find that MPT maintains relatively comparable in most evaluation metrics. It indicates that it is beneficial to leverage diverse prompts to explore the fine-grained features for text-video retrieval. Specifically, our method achieves the best performance on ActivityNet and DiDeMo with R@1 of 41.4% and 46.4%, respectively. In particular, on ActivityNet, the performance of all prompt tuning methods is lower than CLIP4Clip, while MPT outperforms CLIP4Clip with an increase of 0.9%. On LSMDC, MPT also obtains comparable results. The aforementioned results significantly emphasize the advantages of the proposed method.

## 4.4 Ablation Study

In this section, we conduct detailed ablative studies to investigate the impact of the designed components, where all experiments are built on the MPT-Transformer with CLIP-ViT-B/32 backbone.

**Effectiveness of the designed prompts.** In Tab. 3, we investigate the impact of each type of our devised prompt in MPT. Note that we report text-to-video retrieval results (*t2v*) for brevity. Here, the baseline method only exploits basic spatial and word prompts (Line 1). Subsequently, we conduct component-wise analysis on the other prompts (temporal, global, and phrase) by progressively adding them to the baseline method. Overall, all the devised prompts contributed significantly to the overall performance. Specifically, the baseline model first performs the worst. By integrating temporal prompt into baseline (Line 2), the performance obtains larger improvement, particularly increased by 2.0% in R@1. It reveals the importance of temporal information for videos. Then, the global prompt is added to the model (Line 3), which further enhances the

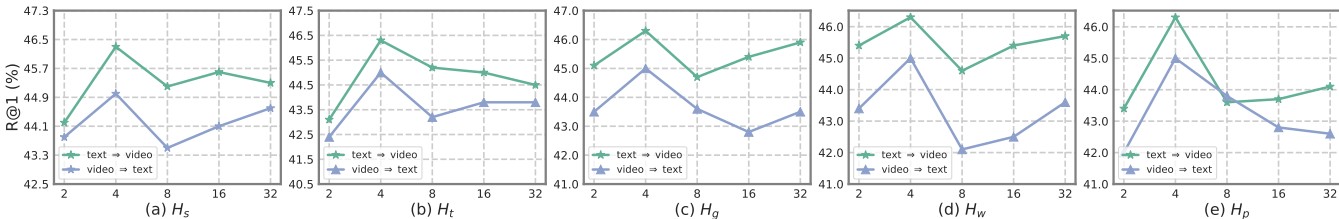

**Figure 3: Ablation study of different prompt lengths. Here, $H_s$, $H_t$, $H_g$, $H_w$, and $H_p$ indicate the length of spatial, temporal, global, word, and phrase prompts respectively.**

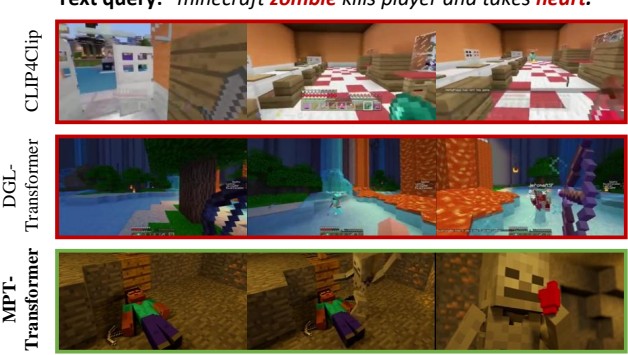

**Figure 4: Qualitative analysis of CLIP4Clip, DGL-Transformer, and our MPT-Transformer with respect to R@1 results on MSR-VTT dataset, where the correct and incorrect videos are highlighted with green and red outlines.**

performance, indicating that the global prompt enables the comprehensive acquirement of video information. Finally, adding the phrase prompt into the model (Line 4) achieves better performance, verifying its efficacy in capturing the semantics of contextualized text. The results show that the proposed prompts are effective in obtaining modality-specific knowledge about the video and text.

**Effectiveness of Prompt Length.** To determine the optimal number of prompt tokens for learning, we conducted experiments with different lengths $H_*$ for five prompts, where $H_* = \{2, 4, 8, 16, 32\}$, as shown in Fig. 3. Empirically, the larger the value of $H_*$, the more contextually relevant knowledge is obtained. However, continuously expanding prompt length does not yield a sustained improvement. The possible reason is that too many prompt tokens may involve redundant noise, harming the valid knowledge acquired from pretrained model. Thus, we experimentally set the number of prompt tokens as 4 for all prompt types in the remaining experiments.

### 4.5 Qualitative Results

In Fig. 4, we showcase some visualization results of our method on the MSR-VTT and compare it with CLIP4Clip and DGL-Transformer. As shown, although CLIP4Clip and DGL-Transformer search out the related videos, they are inferior in capturing multi-grained semantics, failing to recall the ground truth videos. However, MPT-Transformer successfully retrieves correct videos based on text query, recognizing detailed concepts '*zombie*' and '*heart*'. To further

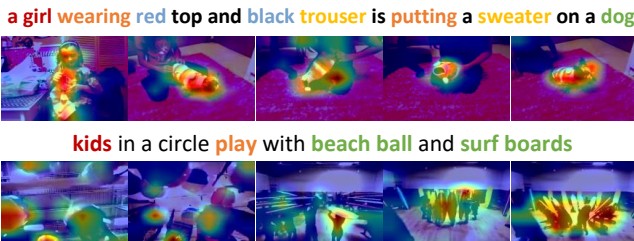

**Figure 5: Qualitative analysis of attention map of global prompt attended to each video frame and clustering results of word embeddings during phrase prompting, where semantic-related words are successfully grouped into the same prototype (marked using the same colors).**

comprehend the knowledge learned by model, we visualize the qualitative results of attention mapping of global prompts and clusterings of phrase embeddings, where the words marked with the same colors indicate they belong to the same centroid. In Fig. 5, we find that: (1) the global prompt captures the discriminative visual clues and dynamics of video. (2) Our clustering algorithm learns the associations among words, *e.g.,* 'wearing' and 'putting', 'red' and 'black' in the top example, and 'beach ball' and 'surfboards' in the bottom example, which contributes to the matching of the two modalities. These examples explicitly illustrate the efficacy of MPT.

## 5 CONCLUSION

In this paper, we study how to effectively and efficiently utilize prompt tuning to acquire domain-specific features when adapting base-generic pre-trained knowledge for text-video retrieval. To achieve this target, we propose MPT, a task-specific prompt tuning that explores the fine-grained features by devising multiple distinctive prompts, focusing on relevant and discriminate knowledge of both modalities. Concretely, MVE employs spatial, temporal, and global prompts to comprehensively capture video-specific features while MTE utilizes word and phrase prompts to learn lexical and contextual semantics. Extensive experiments on the four text-video retrieval benchmarks and visualization analysis prove the effectiveness and interpretability of the proposed method.

## ACKNOWLEDGMENTS

This study is supported by grants from the National Natural Science Foundation of China (Grant No. 62122018, No. 62020106008, No. U22A2097, No. U23A20315), and Kuaishou.

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
