# OpenReview forum: "MPT: Multi-grained Prompt Tuning for Text-Video Retrieval"
_acmmm.org/ACMMM/2024/Conference — MM2024 Poster_

### Official Review · Reviewer_Hm7M · 2024-05-23

**Rating:** 4
**Confidence:** 2

**Summary:**

The paper "MPT: Multi-grained Prompt Tuning for Text-Video Retrieval" proposes a novel approach to text-video retrieval by using a Multi-grained Prompt Tuning (MPT) method. This method aims to enhance the adaptation of large-scale pre-trained image-text models for the text-video retrieval task. The MPT approach involves designing various specific prompts to effectively explore semantic interactions across different modalities with diverse granularity. Specifically, the authors introduce a multi-grained video encoder (MVE) and a multi-grained text encoder (MTE). The MVE incorporates spatial, temporal, and global prompts to capture detailed video-specific characteristics, while the MTE uses word and phrase prompts to capture various levels of textual semantics. Extensive experiments on benchmark datasets (MSR-VTT, ActivityNet, DiDeMo, and LSMDC) demonstrate that MPT achieves outstanding performance, surpassing state-of-the-art methods with minimal computational cost.

**Strengths:**

Strengths
Innovative Multi-grained Approach: The use of multi-grained prompts for both video and text encoders is novel. It effectively captures the diverse semantic characteristics inherent in videos and textual descriptions.

Parameter Efficiency: MPT focuses on parameter-efficient tuning, which involves fewer learnable parameters compared to full fine-tuning methods. This efficiency is crucial for practical deployment and scalability.

Comprehensive Evaluation: The paper provides extensive experiments on multiple benchmark datasets, demonstrating the robustness and generalizability of the proposed method. The results show significant improvements over existing methods.

Detailed Ablation Studies: The inclusion of detailed ablation studies helps in understanding the contributions of different components of the proposed approach. It provides insights into the effectiveness of spatial, temporal, and global prompts in video encoding, as well as word and phrase prompts in text encoding.

**Limitations:**

Limitations
Computational Complexity: While the approach is parameter-efficient, the multi-grained prompt tuning still involves complex processing steps, especially when integrating spatial, temporal, and global prompts. This might limit its feasibility for real-time applications.

Focus on Specific Prompts: The paper focuses on a predefined set of prompts (spatial, temporal, global for video and word, phrase for text). Exploring adaptive prompt generation mechanisms based on the input data characteristics could enhance the model's flexibility and performance.

Scalability Issues: Although the model is parameter-efficient, scaling it to very large datasets or longer video sequences might still pose challenges. Addressing these scalability issues is crucial for broader applicability.

**Suitability:**

3

---

### Official Review · Reviewer_kvyR · 2024-05-24

**Rating:** 3
**Confidence:** 3

**Summary:**

In this work, authors propose a Multi-grained Prompt Tuning method for text-video retrieval. The method contains a variety of specific prompts to effectively explore semantic interaction across different modalities with diverse granularity. They totally use spatial, temporal, and global prompts to fine-tune the model. In addition, multi-grained text encoder and visual encoder are also designed.

**Strengths:**

1. The paper is clearly written and easy to follow.
2. Appropriate examples enhance the understanding of the paper.
3. The method conducts sufficient experiments to prove their conclusions.

**Limitations:**

1. The difference between the temporal prompts and the global prompts is not clear. The temporal prompts and the global prompts both contain the feature of the whole video (all frames). The two items seem semantically redundant.
2. The idea of using prompts and multi-grained prompts is too similar to previous work DGL [42]. The DGL model proposed using local and global prompt tuning methods for the text-video retrieval task, corresponds to the idea of spatial prompts and temporal/global prompts in this work. The usage of linear layers and transformer layers was also first introduced in DGL. From the experimental results in Tables 1 and 2, the improvement in the model's performance is limited compared to the DGL method.
3. The symbol definitions in Fig. 2 are not clear enough, leading to hard understanding.

**Suitability:**

3

---

### Official Review · Reviewer_n1QU · 2024-05-25

**Rating:** 4
**Confidence:** 3

**Summary:**

Previous text-video retrieval methods primarily depend on coarse visual and textual cues, neglecting the fine-grained attributes of videos. Specifically, videos inherently contain multiple visual properties, such as static spatiality (intra-frame), dynamic temporality (inter-frame), and global understanding (whole video), which are often overlooked, leading to sub-optimal performance. To address these issues, this paper introduces Multi-grained Prompt Tuning (MPT) for text-video retrieval, which designs specific prompts to explore semantic interactions across different modalities with diverse granularity. The method incorporates a multi-grained video encoder that employs spatial, temporal, and global prompts to transfer base-generic knowledge while thoroughly exploring video-specific characteristics. Additionally, a novel multi-grained text encoder is introduced to capture various levels of textual clues through the use of word and phrase prompts. Extensive experiments on four benchmark datasets—MSR-VTT, ActivityNet, DiDeMo, and LSMDC—demonstrate that MPT achieves outstanding performance, surpassing state-of-the-art methods with minimal computational cost.

**Strengths:**

(1)	The paper introduces a novel Multi-grained Prompt Tuning (MPT) approach for text-video retrieval, which addresses the limitations of previous methods by effectively capturing fine-grained video properties . The use of specific prompts for different modalities ensures a thorough exploration of video-specific features, enhancing the model's retrieval capabilities.The design includes a multi-grained video encoder employing spatial, temporal, and global prompts to capture fine-grained video characteristics. Additionally, the novel multi-grained text encoder captures various levels of textual clues through word and phrase prompts,which is rational and novel.
(2)	This paper is well-organized, making it easy for readers to follow the logical flow of the research.This clear presentation ensures that the innovative contributions and results are effectively communicated.

**Limitations:**

(1)	Effectiveness of Spatial and Temporal Prompts:The design of spatial and temporal prompts in the MPT method, although innovative compared to the DGL approach, may not be entirely convincing. These prompts introduce additional parameters at the patch embedding and CLS token levels, which could be seen as unnecessary without clear evidence of their substantial impact. The comparative experiments reveal that despite having more tunable parameters, MPT only achieves marginal improvements over DGL, and in some tasks, it even performs worse. This raises concerns about the actual contribution and efficiency of spatial and temporal prompts, as the added complexity does not translate into significant performance gains. A more detailed analysis demonstrating the specific benefits and justifications for these prompts would enhance the credibility of the proposed method.
(2)	Clarity and Consistency in Notation: In the presentation of the spatial and temporal prompts, the paper suffers from issues of clarity and consistency in its notation. Specifically, in Equation (7), the variable j is introduced to represent the j-th frame in a series of input frames. However, this j is not carried forward into the subsequent Equations (8) and (9), leading to potential confusion. Additionally, Figure 2(b) fails to clearly annotate the presence of the j element in the Global Prompt and Temporal Prompt, which further complicates the understanding of these components. This inconsistency in notation can negatively impact the readability and comprehension of the paper, making it difficult for readers to follow the logical flow and understand the detailed workings of the proposed prompts.
(3)	Performance and Ablation Studies: In the experimental comparisons, the proposed MPT method does not consistently outperform the state-of-the-art DGL method. For instance, in the Video-to-Text retrieval task on the MSR-VTT dataset using the ViT-B/16 visual backbone, MPT falls short of DGL in the R@5 and R@10 metrics. This indicates that the improvements offered by MPT are not universally superior across all metrics and settings. Additionally, the paper's ablation studies lack depth, as only four groups of ablations are presented to evaluate the five different prompts proposed. This limited ablation analysis is insufficient to comprehensively demonstrate the individual contributions and effectiveness of each prompt type. More detailed and extensive ablation experiments are necessary to clearly elucidate the specific roles and benefits of each component within the MPT framework.
(4)	Lack of Detailed Review of Prior Work on Multi-Granularity Text-Video Retrieval: The paper does not provide a sufficiently detailed review of prior work on multi-granularity text-video retrieval. To enhance the background and related work sections, the authors should include references to significant research in this area. Recommended papers for citation include:
·Cap4Video: What Can Auxiliary Captions Do for Text-Video Retrieval?, CVPR 2023, which explores the use of auxiliary textual information such as titles, tags, and subtitles to enhance text-video matching. The authors utilize zero-shot captioning models to generate additional textual data, beneficial for fine-grained retrieva.
·UATVR: Uncertainty-Adaptive Text-Video Retrieval, ICCV 2023, which proposes a multi-level representation and matching strategy that comprehensively considers different granularity levels of information in both video and text. The local-global interaction module captures detailed information while maintaining global semantic consistency.
·Align and Prompt: Video-and-Language Pre-training with Entity Prompts, CVPR 2022, which introduces prompt-based entity modeling to learn fine-grained alignment between visual and textual modalities in a self-supervised manner.
·Fine-grained Text-Video Retrieval with Frozen Image Encoders, arXiv 2023, a two-stage text-video retrieval architecture that uses existing text-video retrieval methods for efficient candidate selection, and a decoupled video-text cross-attention module to capture fine-grained multimodal information in spatial and temporal dimensions .
·Fine-grained Video-Text Retrieval with Hierarchical Graph Reasoning, CVPR 2020, which decomposes video-text matching into a global-to-local hierarchical structure, generating hierarchical semantic graphs that include events, actions, and entities, and using attention mechanisms to generate hierarchical textual embeddings.

**Suitability:**

3

---

### Official Review · Reviewer_P3j7 · 2024-05-27

**Rating:** 5
**Confidence:** 3

**Summary:**

This paper presents an efficient method named MPT, which employs multi-grained prompt tuning for text-video retrieval. The multi-grained prompt encompasses global, spatial, and temporal prompts for visual tuning, and incorporates word and phrase prompts for textual tuning. Extensive experiments demonstrate the effectiveness of the MPT method.

**Strengths:**

1. This method considers multi-grained prompt tuning, offering a comprehensive exploration for prompt tuning in the text-video retrieval task.
2. The experiments demonstrate both the effectiveness and efficiency of this method.

**Limitations:**

1. Symbol notation should maintain consistency. While Line 380 defines $F$ as the number of video frames, $F_{i-1}$ indicates the visual prompt for the i-th layer in Line 405. Some subscripts denote layer indices, while others do not. Additionally, the shape of each symbol should be clarified. Is $V_{i-1} \in \mathbb{R}^{M \times D_{v}}$? Does the $c_{i}$ represent the [CLS] token?
2. The method description should be more detailed. For example, in equation (7), the shape of $P_{i-1,j}^{S}$ is $\mathbb{R}^{H_S \times D_{v}}$, how does it concatenate with the $V_{i-1} \in \mathbb{R}^{M \times D_{v}}$? Which dimension is processed in the encoder?
3. Regarding Table 3, the performance of global prompts seems suboptimal for retrieval, as the R@5 doesn't increase, while MnR increases. What's the performance when only ablating the global prompts?
4. How does the model perform when incorporating sentence prompts, i.e., utilizing the first token as the sentence-level feature?

**Suitability:**

3

---

### Meta-Review · Area_Chair_7rgK · 2024-06-27

**Recommendation:** Accept (Poster)
**Confidence:** 4

**Metareview:**

This submission initially received relatively positive reviews: 1 Weak Accept, 2 Borderline Accept, and 1 Borderline Reject. The main concerns are about the: 1) Novelty of the proposed method (especially the comparison with prior work DGL). 2) The presentation needs to be further improved. 3) Generalization ability of the proposed method. After the rebuttal, the main concerns were addressed, and one reviewer raised the rating to Weak Accept. Overall, we think this submission is worth to be published with significant contributions to the community. Thus, we recommend Accept.